# Role of microRNA-21 and Its Underlying Mechanisms in Inflammatory Responses in Diabetic Wounds

**DOI:** 10.3390/ijms21093328

**Published:** 2020-05-08

**Authors:** Cole Liechty, Junyi Hu, Liping Zhang, Kenneth W. Liechty, Junwang Xu

**Affiliations:** Laboratory for Fetal and Regenerative Biology, Department of Surgery, University of Colorado Denver-Anschutz Medical Campus and Children’s Hospital Colorado, Aurora, CO 80045, USA; cole.liechty@cuanschutz.edu (C.L.); junyi.hu@cuanschutz.edu (J.H.); liping.2.zhang@cuanschutz.edu (L.Z.); ken.liechty@CUanschutz.edu (K.W.L.)

**Keywords:** diabetic wounds, microRNA, miR-21, macrophage polarization, ROS

## Abstract

A central feature of diabetic wounds is the persistence of chronic inflammation, which is partly due to the prolonged presence of pro-inflammatory (M1) macrophages in diabetic wounds. Persistence of the M1 macrophage phenotype and failure to transition to the regenerative or pro-remodeling (M2) macrophage phenotype plays an indispensable role in diabetic wound impairment; however, the mechanism underlying this relationship remains unclear. Recently, microRNAs have been shown to provide an additional layer of regulation of gene expression. In particular, microRNA-21 (miR-21) is essential for an inflammatory immune response. We hypothesize that miR-21 plays a role in regulating inflammation by promoting M1 macrophage polarization and the production of reactive oxygen species (ROS). To test our hypothesis, we employed an in vivo mouse skin wound model in conjunction with an in vitro mouse model to assess miR-21 expression and macrophage polarization. First, we found that miR-21 exhibits a distinct expression pattern in each phase of healing in diabetic wounds. MiR-21 abundance was higher during early and late phases of wound repair in diabetic wounds, while it was significantly lower in the middle phase of wounding (at days 3 and 7 following wounding). In macrophage cells, M1 polarized macrophages exhibited an upregulation of miR-21, as well as the M1 and pro-inflammatory markers IL-1b, TNFa, iNos, IL-6, and IL-8. Overexpression of miR-21 in macrophage cells resulted in an upregulation of miR-21 and also increased expression of the M1 markers IL-1b, TNFa, iNos, and IL-6. Furthermore, hyperglycemia induced NOX2 expression and ROS production through the HG/miR-21/PI3K/NOX2/ROS signaling cascade. These findings provide evidence that miR-21 is involved in the regulation of inflammation. Dysregulation of miR-21 may explain the abnormal inflammation and persistent M1 macrophage polarization seen in diabetic wounds.

## 1. Introduction

Diabetes has reached pandemic proportions worldwide. Expenditures on diabetic care in the US alone surpassed USD 327 billion in 2017 (data from the 2017 National Diabetes Statistics Report and www.diabetes.org). Complications of diabetes, such as impaired wound healing, represent a significant medical problem. The annual cost of treating diabetic lower extremity ulcers alone exceeds USD 1.5 billion [1], further highlighting the magnitude of this problem. In addition, an ulcer of the lower extremity precedes 84% of all diabetic lower extremity amputations and is the primary cause for hospitalization among diabetics [2]. Despite the enormous impact of these wounds on both individuals and the economy, effective therapies remain elusive. Thus, the treatment or correction of diabetes impaired wound healing has far-reaching implications, both on patient outcomes as well as on healthcare expenditures.

A central pathogenic feature of diabetic wounds is the persistence of chronic inflammation [3], partly due to the prolonged existence of pro-inflammatory macrophages, known as pro-inflammatory (M1) macrophages [4,5,6] Macrophages are essential in all phases of wound healing. Early in the response to injury, macrophages are polarized to the M1 phenotype and produce pro-inflammatory cytokines that stimulate the production of reactive oxygen species (ROS) to promote clearance of bacteria and debris from the wound [5,6]. After the initial inflammatory phase, macrophages transition to an pro-remodeling (M2) macrophage phenotype, which is associated with the resolution of the inflammatory response and promotion of wound remodeling and closure [5,6]. In diabetic wounds, the persistence of the M1 macrophage phenotype and a failure to transition to the M2 phenotype is associated with delayed healing [7]; however, the underlying mechanism is not clear.

The role of microRNAs in the regulation of macrophage plasticity and polarization has been increasingly studied [8]. MicroRNA miR-155 was highly induced in M1-polarized macrophages [9], while miR-146a expression [10,11] was elevated in M2-polarizing conditions, demonstrating the variable roles that different microRNAs play in the wound healing process. Multiple regulatory roles of miR-21 in wound healing have been reported [12,13,14,15,16]. MiR-21 was also associated with hyperglycemia and ROS formation by regulating SOD2 in HUVECs [17], demonstrating that circulating miR-21 can also serve as an early predictor of developing diabetes [18]. However, the role of miR-21 in macrophage polarization, which is closely related to its role in the inflammatory response, is not clear and remains controversial [19,20,21].

Thus, we hypothesize that miR-21 is dysregulated in diabetic wounds, and dysregulated miR-21 may partly be responsible for the prolonged M1 macrophage phenotype seen in diabetic wounds. In the inflammatory phase of wound healing, inflammatory cells are activated and recruited to the wound area, producing large amounts of ROS, matrix metalloproteinases, and pro-inflammatory cytokines [22,23]. These inflammatory cells express high levels of NOX2 and produce high levels of ROS [24,25], crucial to the neutralization of invading pathogens. Therefore, we further hypothesize that miR-21 promotes ROS production through NOX2 regulation via the PI3K pathway in macrophages.

## 2. Results

### 2.1. Dynamic miR-21 Expression in Diabetic Wounds

At baseline, diabetic and non-diabetic skin (day 0) exhibited similar levels of miRNA-21 RNA abundance (Figure 1). In response to injury, miR-21 expression was induced in both non-diabetic (non-Db) and diabetic (Db) wounds. Interestingly, the miR-21 expression pattern after the injury can be separated into 3 phases. At the early phase of the injury, we found that miR-21 RNA abundance was significantly higher in diabetic wounds compared to non-diabetic wounds at day 1 following wounding. In the middle phase, at days 3 and 7 following wounding, miR-21 RNA abundance was significantly lower in diabetic wounds compared to non-diabetic wounds. However, in the end phase, at days 14 and 21 following wounding, miR-21 RNA abundance was significantly higher in diabetic wounds, while in non-Db wounds, miR-21 was gradually reduced to the level before wounding. This interesting expression pattern of miR-21 suggests that miR-21 has multiple functions in diabetic wounds. In Figure 1B, we further confirmed that miR-21 RNA abundance was higher in diabetic wounds on day 1 following injury (*n* = 5 per group). We isolated wound macrophage at day 1 following wounding. The abundance of miR-21 was significantly higher in diabetic wound macrophage compared to non-Db day 1 wound macrophage (Figure 1C). We also examined the abundance of miR-21 in human non-diabetic and diabetic skin. Similar to the situation of mouse diabetic wounds at day 1, miR-21 expression was significantly upregulated in human diabetic skin compared to human non-diabetic skin (Figure 1D).

### 2.2. miR-21 Is Significantly Induced in M1 Macrophage Cells

To understand the higher expression level of miR-21 in the early phase of diabetic wound healing, we did an analysis of its expression in macrophages. We asked whether miR-21 was differentially expressed in the M1 and M2 macrophage phenotypes. First, we induced RAW 264.7 murine macrophages with lipopolysaccharide (LPS) (10 pg/mL) and IFN-r (20 ng/mL) for 24 h to generate M1 macrophages, or IL-4 (20 ng/mL) for 24 h to generate M2 macrophages after overnight serum starvation. To confirm the macrophage phenotype after treatment, we measured the expression of M1 or M2 marker genes following the different treatments. Figure 2 shows that LPS + IFN-r treated macrophages were polarized to the M1 macrophage phenotype based on the expression of M1 marker genes (iNOS, IL1 beta, and TNFa) (Figure 2A–D), while IL-4 treated macrophages were polarized to the M2 phenotype based on the expression of M2 marker genes (Arg1 and Mrc1; Figure 2E,F). This data confirmed the macrophages were correctly polarized to the M1 and M2 phenotypes following their respective treatment.

Next, we measured the miR-21 RNA abundance in different macrophage phenotypes. We tested the relationship between LPS and miR-21 expression. Similar to other reports [26,27], we also confirmed that LPS induces miR-21 expression in a dose- (Figure 3A) and time-dependent (Figure 3B) manner. Real-time PCR analysis on the confirmed polarized macrophages further indicated that miR-21 was highly expressed in M1 macrophages (Figure 3C).

### 2.3. Overexpression of miR-21 Induces the Pro-Inflammatory Macrophage Phenotype Verified through Increased M1 Marker mRNA Abundance

To test whether miR-21 may play a role in macrophage polarization, RAW macrophages were transfected with miR-21 mimic, or mimic control. Our results showed that miR-21 RNA abundance was increased in the mimic-transfected cells, which confirmed the successful transfection of the RAW cells with miR-21 mimic (Figure 4A). Next, we measured M1 or M2 marker genes expression following transfection with either miR-21 mimic or mimic control. Figure 4 showed that miR-21 overexpression further induced mRNA abundance of M1 marker genes (iNOS, IL1beta, and TNFa) compared to RAW macrophages treated with the mimic control (Figure 4B–E). This data suggests that miR-21 is a positive inducer for pro-inflammatory macrophages.

### 2.4. Hyperglycemia Induces miR-21 and Reduces pTEN Expression in Macrophage Cells

Since diabetic-wound-derived cells are consistently under hyperglycemic conditions, we asked whether hyperglycemia can alter miR-21 expression in macrophages. After the culture of RAW macrophages in low (5 mM) or high (25 mM) glucose media for 4, 8, or 24 h, real-time qPCR demonstrated that miR-21 was significantly induced by hyperglycemia in a time-dependent manner (Figure 5A). Phosphatase and tensin homolog (pTEN) is one of the direct targets of miR-21 [28,29], prompting us to ask whether hyperglycemia will alter pTEN expression. RNA abundance analysis indicated that pTEN was significantly reduced by hyperglycemia in a time-dependent manner, inversely associated with miR-21 expression (Figure 5B). In order to establish the causal relationship between miR-21 and pTEN, we analyzed the mRNA abundance of pTEN in gain and loss of miR-21 expression. The data indicated that overexpression of miR-21 by mimic transfection significantly reduced PTEN mRNA abundance (Figure 5C), while knockdown of miR-21 by antagomir transfection significantly induced PTEN mRNA abundance (Figure 5C).

### 2.5. NOX2 Is Induced by miR-21 through PI3K Resulting in ROS Production

In our previous studies, we reported that hyperglycemia significantly induces the production of ROS in RAW cells, and higher expression (2 fold higher in high glucose condition) of the NOX2 gene was responsible for higher ROS production [30]. In this study, we confirmed that NOX2 mRNA abundance was highly induced under hyperglycemic conditions. To further determine whether miR-21 targets NOX2 in macrophages, RAW cells were transfected with miR-21 mimic, miR-21 inhibitor, or their control. First, we found that miR-21-mimic-treated RAW cells showed significantly higher ROS production compared to control-mimic-treated RAW cells (Figure 5D). As shown in Figure 5E, overexpression of miR-21 significantly induced NOX2 mRNA abundance. Conversely, inhibition of miR-21 significantly reduced NOX2 mRNA abundance (Figure 5F). This data indicates that miR-21 expression is inversely related to NOX2 gene expression.

We then asked how miR-21 was responsible for NOX2 expression. PTEN has been shown to be a direct target of miR-21 [25,26]. Its expression can act to downregulate the phosphoinositide 3-kinase (PI3K) pathway, while the loss of PTEN expression induces the PI3K pathway [25,26]. The PI3K pathway has been confirmed throughout multiple studies to positively modulate NOX2 expression in many cells. We evaluated NOX2 and PI3K protein expression by Western blot. We found that NOX2 and pPI3K were significantly induced by miR-21 overexpression compared to mimic control (Figure 6A). We also found a significant increase in expression of pERK, a key component in the MEK/ERK signaling pathway, and an important player in cross-talk with the PI3K pathway (Figure 6A). Densitometry analysis was presented in Figure 6B, also indicating that overexpression of miR-21 significantly induced the expression of pI3K, pERK, and NOX2. These findings suggest that miR-21 was a positive regulator of PI3K and NOX2 expression.

To test whether miR-21 regulates NOX2 through the PI3K pathway, RAW cells were pre-treated with either DMSO or 10 µM Ly294002 (inhibitors of PI3K activity) for 30 min, then treated with low (5 mM) or high (25 mM) glucose media for 24 h. As seen in Figure 6C, hyperglycemia significantly induced NOX2 expression in pre-treated RAW cells with DMSO. However, when RAW cells were pre-treated with inhibitor Ly294002, they exhibited a significant reduction of NOX2 mRNA abundance. This data indicates that the PI3k pathway plays a critical role in NOX2 expression under hyperglycemic conditions. This data also suggests that hyperglycemia-induced NOX2 expression and ROS production may be done through the HG/miR-21/PI3K/NOX2/ROS signaling cascade (Figure 6D).

## 3. Discussion

Normal wound repair follows an orderly and well-defined sequence of events that requires the interaction of many cell types and growth factors, which are divided into the inflammatory, proliferative, and remodeling phases [31]. In diabetic wound healing, this complex orchestration of wound healing processes and phases are disrupted [32]. Recently, dysregulated microRNAs which have been reported in diabetic wound healing, highlighting the multifactorial roles of microRNAs in the wound healing process [33,34,35,36]. MiR-21 is a highly studied microRNA, and its function in wound healing has been implicated in inflammatory, proliferative, and remodeling phases, throughout different cells [13,14,15,16,21,29,37]. MiR-21 has been reported as a microRNA with anti-inflammatory, anti-apoptotic, and pro-proliferative properties, and has been found to play a role in regulating wound contraction and collagen deposition [13,14,15,16,21,29,37]. Increasing evidence indicates that miR-21 may be involved in macrophage polarization. Xi et al. indicated that miR-21 is responsible for M1 macrophage polarization by targeting STAT1 [38]; however, another report suggested that miR-21 plays no role in M1 phenotype and the target for miR-21 is STAT3, which are crucial for the M2 macrophage phenotype [15]. It will be informative to look into the role of miR-21 in macrophage polarization. In the present study, we compared miR-21 expression between non-diabetic wounds and diabetic wounds and found that miR-21 was differentially expressed among these conditions. More interestingly, its expression can be classified into 3 phases: miR-21 was higher in diabetic wounds at day 1 after injury (inflammatory phase), then at days 3 and 7 after injury (proliferative phase), miR-21 was lower in diabetic wounds, then at days 14 and 21 after injury (remodeling phase), miR-21 was significantly higher in diabetic wounds. Furthermore, we confirmed that miR-21 was higher on day 1 after wounding in diabetic wounds, diabetic wound macrophage, and in human diabetic skin. The data indicates miR-21 may play different roles in each phase of the wound healing process, also implying the importance of time on the modulation of miR-21 expression in its target therapy.

Macrophages have been shown to play a critical role in the wound healing processes [8,39]. Macrophages can adapt to different phenotypes (M1 or M2) in response to environmental stimuli. During acute inflammation, classically activated macrophages (M1) predominate, while during the resolution phase, alternative macrophages (M2) are dominant [5,6]. In diabetic wounds, the persistence of the M1 macrophage phenotype and a failure to transition to the M2 phenotype is associated with delayed healing [7]. In this study, we induced macrophages toward M1 or M2 phenotypes by well-defined factors, and phenotypes were confirmed by M1 or M2 marker gene expression analysis. We found that miR-21 was significantly induced in M1 macrophages. Furthermore, the exogenous addition of miR-21 with transfection of miR-21 mimic can further induce M1 marker genes expression, indicating miR-21 may be involved in M1 macrophage polarization.

Another important finding is that hyperglycemia induces miR-21 expression, which represses pTEN, one of the direct targets of miR-21. Downregulated pTEN indirectly induces NOX2 and ROS production. ROS production is critical for the activation and function of M1 macrophages [40]. Hyperglycemia has been described as an inducer for ROS production [41]. In this study, we added miR-21 to this signaling cascade and found that miR-21 can be induced by high glucose in a time-dependent manner, inversely reducing its target gene pTEN. PTEN has been demonstrated to be a target of miR-21 throughout several studies [28,29]. The inhibition of pTEN can lead to the activation of PI3K, which can induce NOX2 genes. We inhibited PI3K with LY294002, and induction of NOX2 by hyperglycemia was attenuated with PI3K inhibition, indicating that miR-21 induces NOX2 expression through the PI3K pathway.

In summary, these findings provide promising evidence that miR-21 is involved in the regulation of macrophage polarization, ROS production, and modulating inflammatory responses, implicating a role for miR-21 in the early phase of diabetic wound healing. Furthermore, these results suggest a potential role of miR-21 in the pathogenesis of diabetic wound healing impairment and implicate it as a novel therapeutic target to correct this impairment.

## 4. Materials and Methods

### 4.1. Animal Studies

Animal experiments were approved by the Institutional Animal Care and Use Committee at the University of Colorado Denver–Anschutz Medical Campus and performed in accordance with relevant guidelines and regulations (protocol 00186, approval date 14 August 2019). In these experiments, we used 10 weeks old age-matched, female, genetically diabetic C57BKS.Cg-m/Leprdb/J (Db) mice and heterozygous, non-diabetic (non-Db), female controls were obtained from the Jackson Laboratory (Bar Harbor, ME, USA) and used in this experiments. Mice were anesthetized with inhaled isofluorane. Each mouse was shaved and depilated before wounding. The dorsal skin was swabbed with alcohol and Betadine (Purdue Pharma, Stamford, CT, USA). Each mouse underwent a single, dorsal, full-thickness wound (including panniculus carnosum) with an 8-mm punch biopsy (Miltex Inc, York, PA, USA). All wounds were dressed with tegaderm (3M, St Paul, MN, USA), which was subsequently removed on postoperative day 2. Postoperatively, the mice received a subcutaneous injection of an analgesic, Banamine (Schering-Plough Animal Health Corp., Union, NJ, USA). A full-thickness skin sample, centered on the wound, was harvested 0, 1, 3, 7, 14, and 21 days after surgery (*n* = 5 per time point).

### 4.2. Human Samples

This study was approved through the institutional review board at the University of Colorado Denver–Anschutz Medical Campus (IRB protocol 14-1758, approval date 10/07/2014) and performed according to the relevant guidelines and regulations. Informed family consent was obtained from all the patients included in this study. Human skin samples were collected postmortem from the anterior portion of the lower extremity of individuals with and without diabetes. Samples were obtained from patients who were 45 to 65 years of age and did not have any known comorbid malignancy or a history of radiation or chemotherapy. The skin samples were immediately flash-frozen in liquid nitrogen.

### 4.3. Cell Culture and Reagents

RAW 264.7 macrophages (ATCC, Manassas, VA, USA) were maintained in DMEM (Gibco, Waltham, MA, USA) supplemented with 10% (*v*/*v*) FBS (Gibco, Waltham, MA, USA), 100 μg/mL streptomycin, and 100 U/mL penicillin, and incubated at 37 °C and an atmosphere of 5% CO2. Cells were serum-starved overnight in DMEM with 1% FBS and 100 μg/mL streptomycin, and 100 U/mL penicillin before being treated with 5 mM D-glucose (LG) DMEM, 25 mM D-glucose (HG), for 4, 8, 24 h according to [42]. The pharmacological inhibitors specific to PI3K, LY294002, were purchased from Selleckchem and were applied at 10 uM for 30 min before glucose treatment. RAW cells in this study were analyzed at passage 2 to 3. Wound macrophages will be harvested as described [43], 10^5^ wound macrophages can be routinely obtained per 100 mg wounds, and we have confirmed that these cells are ~98% macrophages based on F4/80 and CD11b staining.

### 4.4. miR-21 Expression Modulation and Transfection

For overexpression, macrophages were transfected with miR-21 mimic or mimic control. For inhibition of miR-21, macrophages were transfected with appropriate target anti-miR-21 antagomir or anti-miR-negative. Transfection reagents, mimic, antagomirs, and control miRNAs were purchased from Invitrogen. After 2 h transfection for RAW cells, the medium was replaced with fresh DMEM supplemented with 10% (*v*/*v*) FBS, 100 μg/mL streptomycin, and 100 U/mL penicillin. Then, 24 h following transfection, the cells were processed for gene expression analysis.

### 4.5. Western Blot Analysis

Cell lysates were prepared in standard NP-40 lysis buffer (Abcam, USA) supplemented with proteinase and phosphatase inhibitors. Protein lysates were quantified using the Pierce BCA protein assay kit (Thermo Fisher Scientific, Waltham, MA, USA). Equal masses of total protein were separated on 4–12% SDS-polyacrylamide mini-gels and blotted onto PVDF membranes (Millipore, Burlington, MA, USA). Membranes were subsequently blocked, incubated with primary antibodies, and incubated with secondary antibodies, according to WesternBreeze Chromogenic Kit (Thermo Scientific, Waltham, MA, USA). Alkaline phosphatase was detected on the PVDF membranes using a ready-to-use BCIP/NBT substrate (Thermo Scientific, USA) for ready visualization of enzyme-linked antibodies. Rabbit anti-PI3K, anti-phospho-PI3K, anti-ERK, anti-phospho-ERK, and anti-GAPDH antibodies were obtained from Cell Signaling (USA). Quantification of relative intensities was achieved by ImageJ analysis.

### 4.6. Intracellular ROS Measurement

After the RAW cells reached 70% confluence, they were subjected to transfection before being exposed for 24 h to normal and high glucose conditions in an FBS-free medium. Intracellular production of hydroxyl, peroxyl, and other ROS was measured by the Cellular Reactive Oxygen Species Detection Assay Kit (Abcam, Cambridge, MA, USA). After transfection, the RAW cells were exposed to 2′,7′-dichlorofluorescein diacetate (DCFDA) for 20 min. The level of intracellular ROS was assessed by the fluorescence emitted by DCFDA after conversion to 2′,7′-dichlorofluorescein by reaction with ROS. The excitation and emission wavelengths were 492 and 521 nm, respectively; ROS levels were recorded by arbitrary unit.

### 4.7. Real-Time Quantitative PCR

Total RNA was extracted with TRIzol reagent (Invitrogen, Carlsbad, CA, USA) according to the manufacturer’s established protocol. Quantitative RT–PCR analyses for miR-21 and RNU6 (used as a normalization control) were performed using TaqMan miRNA assays with reagents, primers, and probes obtained from Qiagen. For RNA abundance analysis, RNA was converted into cDNA using the SuperScript First-Strand Synthesis System (Invitrogen, Carlsbad, CA, USA). IL-1b, TNFa, iNos, IL-6, PTEN, and NOX2 were amplified using the TaqMan gene expression assay (Applied Biosystems, Waltham, MA, USA). Internal normalization was achieved by using the GAPDH housekeeping gene. Samples (*n* = 5 per group) were amplified in triplicate, and results were averaged for each individual sample. The ΔΔCT method was used to calculate relative gene expression. Results are reported as mean ± SD.

### 4.8. Statistical Analysis

Results are expressed as mean ± SD for *n* = 3 to 5 number of independent experiments. Statistically significant differences in gene expression between the two groups were assessed by Student’s *t*-test; *p* < 0.05 was considered to be statistically significant.

## 5. Conclusions

In this study, we provided the dynamic expression pattern of miR-21 during diabetic wound healing. In the early phase of diabetic wound healing, miR-21 may involve in ROS production, induction of pro-inflammatory macrophage phenotype, and inflammatory response. The dynamic expression of miR-21 also indicates the importance of the time of miR-21 based therapy on diabetic wound healing. 

## Figures and Tables

**Figure 1 ijms-21-03328-f001:**
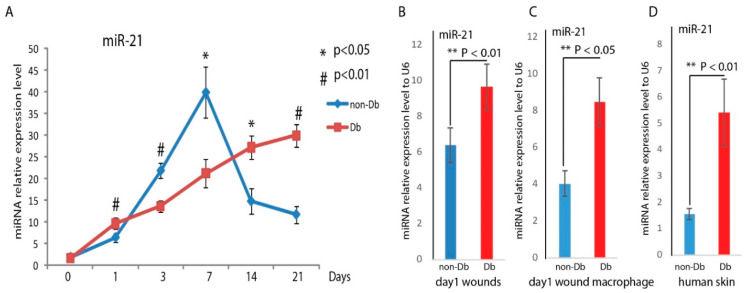
Dynamic RNA abundance changes of miR-21 during the wound healing process. (**A**) Real-time qPCR analysis of miR-21 levels in diabetic (Db)/+ (*n* = 5) and Db/Db (*n* = 5) wounds at days 0, 1, 3, 7, 14, and 21 after dermal injury. MiR-21 RNA abundance was calculated after normalizing with U6. * *pp* < 0.01 comparing Db/Db wounds to Db/+ wounds; (**B**) RNA analyses by real-time qPCR showed significantly increased miR-21 RNA abundance in mouse diabetic and non-diabetic dermal wounds at day 1 (mean+ SD, *n* = 5 per group) after injury, also in diabetic day 1 wound macrophage (**C**), and (**D**) in human non-diabetic and diabetic skin.

**Figure 2 ijms-21-03328-f002:**
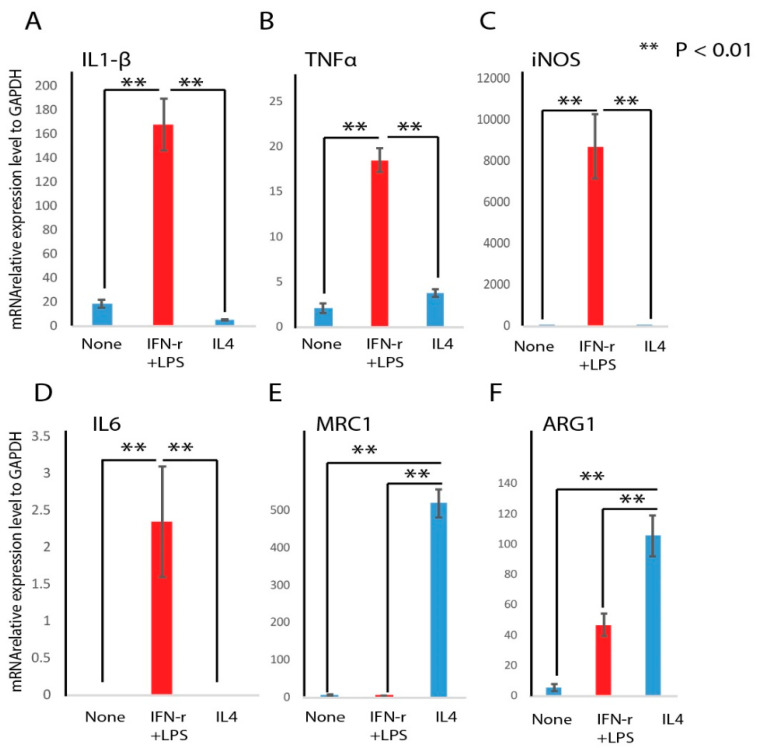
Pro-inflammatory (M1) and regenerative or pro-remodeling (M2) marker mRNA abundance analysis after induction. RAW macrophages were treated with LPS and IFN-r or IL4 for 24 h to induce the M1 or M2 phenotype, respectively. M1 marker genes IL1-beta (**A**), TNFa (**B**), iNOS (**C**), and IL6 (**D**), or M2 marker genes MRC1 (**E**) and ARG1 (**F**) were determined by real-time PCR (*n* = 3, mean + SD, ** *p* < 0.01).

**Figure 3 ijms-21-03328-f003:**
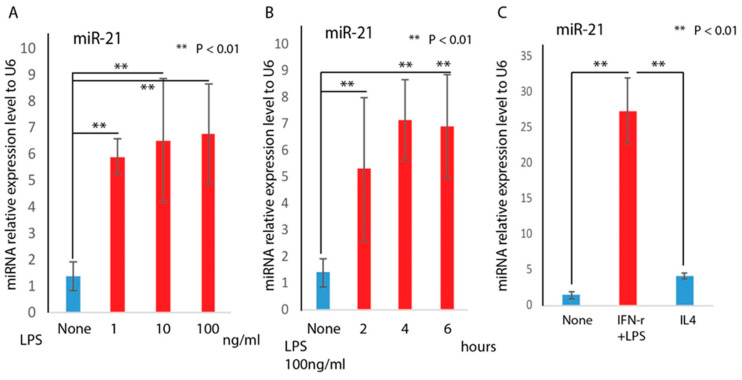
MiR-21 is highly induced in M1 macrophage. RAW cells were treated with LPS at 1, 10, and 100 ng/mL and compared to non-treated RAW cells. RAW cells were also treated with LPS at 100 ng/mL for 2, 4, and 6 h. MiR-21 was significantly induced by LPS in a dose- (**A**) and time-dependent (**B**) manner confirmed by real-time PCR analysis (mean + SD, *n* = 3 per group); (**C**) RNA analyses by real-time qPCR (mean + SD, *n* = 3 per group) indicated that miR-21 was significantly higher in M1 macrophages compared to the untreated control (** *p* < 0.01), and IL4-treated group (** *p* < 0.01).

**Figure 4 ijms-21-03328-f004:**
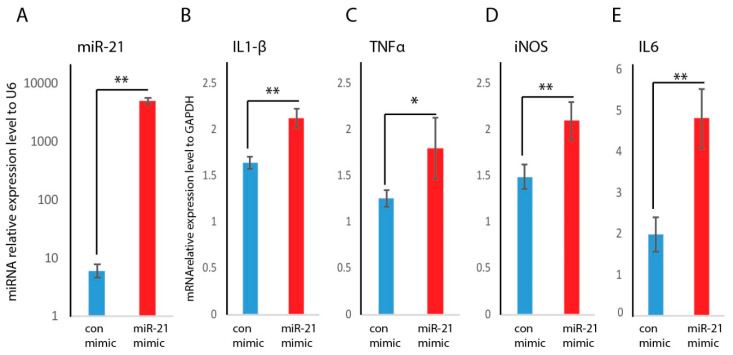
Effects of overexpression of miR-21 on the induction of the pro-inflammatory macrophage phenotype and expression of M1 marker genes. (**A**) Overexpression of miR-21 RNA abundance was achieved by miR-21 mimic transfection and confirmed by RNA analyses by RT–qPCR (mean + SD, *n* = 3 per group), miR-21 abundance was significantly induced in the RAW macrophages transfected with miR-21 mimic compared to mimic control. M1 marker genes IL1-beta (**B**), TNFa (**C**), iNOS (**D**), and IL6 (**E**) were induced following transfection of RAW macrophages with miR-21 mimic (mean + SD, *n* = 5 per group, (** *p* < 0.01, * *p* < 0.05).

**Figure 5 ijms-21-03328-f005:**
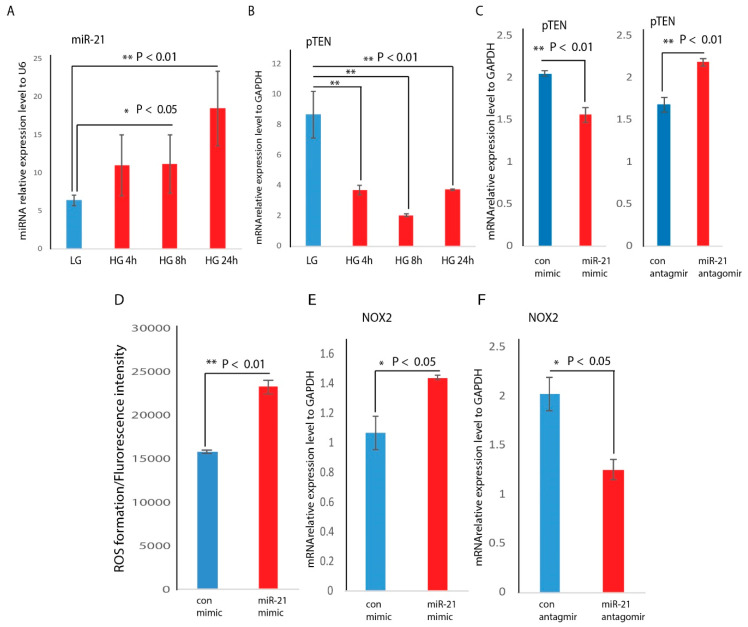
Hyperglycemia induces miR-21 and reduces pTEN expression in macrophage cells. (**A**) RAW cells were treated with high glucose conditions (25 mM D-glucose) for 4, 8, and 24 h. Compared to normal glucose treated RAW cells, miR-21 was significantly induced by high glucose (HG) in a time-dependent manner by real-time PCR analysis (mean + SD, *n* = 3 per group); (**B**) RAW cells were treated with high glucose conditions (25 mM D-glucose) for 4, 8, and 24 h. Compared to normal glucose-treated RAW cells, pTEN was significantly reduced by HG in a time-dependent manner (mean + SD, *n* = 3 per group). Overexpression or knockdown of miR-21 RNA abundance was achieved by miR-21 mimic or antagomir transfection in macrophage; (**C**) real-time PCR analysis on pTEN mRNA abundance under the condition of overexpression or knockdown of miR-21 RNA abundance; (**D**) ROS production between control-mimic- and miR-21-mimic-treated RAW cells using a DCFDA/H2DCFDA-cellular ROS assay. The data is represented as mean ± SD, *n* = 5 per treatment group. NOX2 was significantly induced by miR-21 mimic (**E**), and reduced by miR-21 antagomir (**F**) by real-time PCR analysis (mean + SD, *n* = 3 per group, * *p* < 0.05).

**Figure 6 ijms-21-03328-f006:**
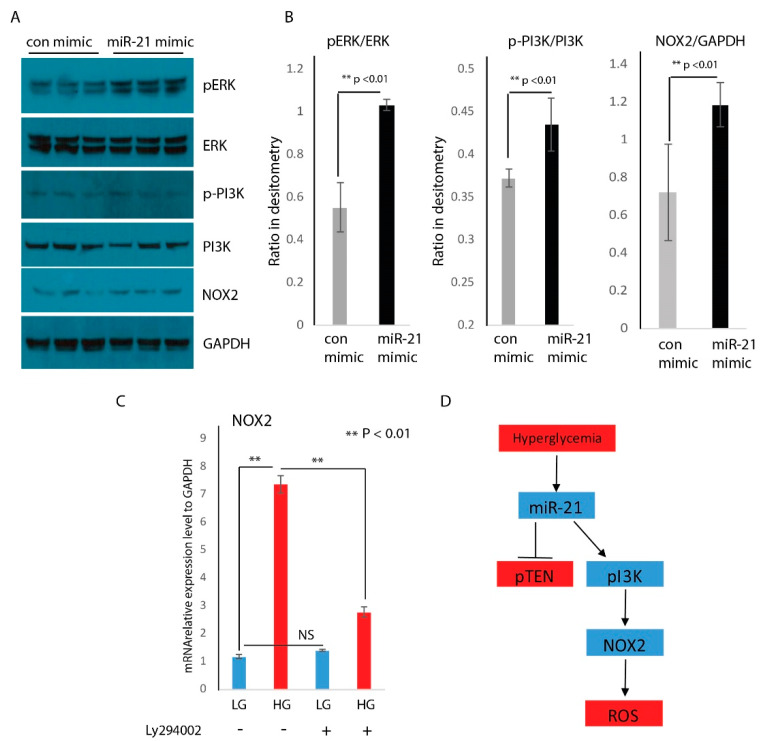
NOX2 is induced by hyperglycemia, and its induction is dependent on the PI3K pathway. (**A**) Western blot analysis of pERK, ERK, pPI3K, PI3K, and NOX2 following transfection of RAW macrophages with miR-21 mimic or control mimic. GAPDH served as internal loading control; (**B**) quantification of relative intensities for the ratio of pERK/ERK, pPI3K/PI3K, and NOX2/GAPDH from Western blot; (**C**) real-time PCR analysis indicated that NOX2 mRNA abundance was induced by hyperglycemia and attenuated by Ly294002 treatment to inhibit the PI3K pathway (mean + SD, *n* = 3 per group, ** *p* < 0.01); (**D**) proposed model of miR-21 and its induction of NOX2 and ROS production.

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
