# Peer review of "Role of microRNA-21 and Its Underlying Mechanisms in Inflammatory Responses in Diabetic Wounds"

_ijms, 2020, doi:10.3390/ijms21093328_

Round 1
Reviewer 1 Report
Dear Authors
the manuscript entitled "microRNA-21 regulates macrophage polarization and ROS production" would demonstrate the role of this miR in the regulation of inflammation in diabetic wound healing. The authors found that miR-21 is induced in M1 polarization, inducing some pro-inflammatory genes.
Further, the authors hypothesize a strong effect of miR-21 on ROS formation by activation of the HG/miR-21/PI3K/NOX2/ROS axis.
The manuscript appears well written and fits with the high-content of the journal.
As reported by La Sala et al. (DOI: 10.1186/s12933-018-0748-2; DOI: 10.1186/s12933-019-0824-2), miR21 has been strictly associated with hyperglycemia and ROS formation. Please, quote in the manuscript these two recent papers regarding ROS formation/miR-21 axis.
Major points:
The authors speculated that ROS are induced by NOX2 (that in hyperglycemia increased). In the figures, any data regarding ROS formation was shown. Please, it is strongly suggested add these results to conclude that the axis HG/miR-21/PI3K/NOX2/ROS is involved in the cascade of ROS production.
- NOX2 protein expression is lacking. A western blot with mimic-miR21 or anti miR21 would be sufficient to demonstrate its involvement in this process;
- the authors should measure ROS levels in cells.
- change "con mimic" with NT (not treated).
Author Response
Reviewer 1:
As reported by La Sala et al. (DOI: 10.1186/s12933-018-0748-2; DOI: 10.1186/s12933-019-0824-2), miR21 has been strictly associated with hyperglycemia and ROS formation. Please, quote in the manuscript these two recent papers regarding ROS formation/miR-21 axis.
- Thank you for providing these important references. In the introduction, we highlighted the importance of these works and cited the references (line 57-59).
The authors speculated that ROS are induced by NOX2 (that in hyperglycemia increased). In the figures, any data regarding ROS formation was shown. Please, it is strongly suggested add these results to conclude that the axis HG/miR-21/PI3K/NOX2/ROS is involved in the cascade of ROS production.
- Thank for these comments. We measured ROS production between control mimic and miR-21 mimic treated RAW cells using DCFDA/H2DCFDA-cellular ROS Assay, miR-21 mimic treated RAW cells showed significantly higher ROS production compared to control mimic treated RAW cells. We added this data on Figure 5, panel C.
- NOX2 protein expression is lacking. A western blot with mimic-miR21 or anti miR21 would be sufficient to demonstrate its involvement in this process;
- Great point. We added western blot data for NOX2 protein level comparing control mimic treated and miR-21 mimic treated RAW cell samples. Densitometry analysis indicated that NOX2 was highly induced by overexpression of miR-21.
- change "con mimic" with NT (not treated).
- Thank you for this suggestion. We feel that NT means not treated, however, con mimic means the cells was treated with control mimic. Since we also had treated with control antagmiR, we think to keep con mimic. It should be more accurate and informative.

Reviewer 2 Report
Liechty and colleagues showed a pro-M1 polarazing role for miR-21, along with a sustained, long-lasting expression in diabetic wounds for this miR in db mice and also its overexpression in human diabetic wounds.
These findings are not novel since a key role for mir 21 in diabetic wounds has been shown (ref 14-18), as has a role in macrophage polarization (Xi et al 2018 oncogene, Wang et al., 2015 Plos one; Sahraei et al., 2019 JCI). Similarly, the increase of miR-21 in the diabetic envirnoment (human, mouse, intracellular and extracellular) has been largely described (Prattichizzo et al., 2018 Redox Biol; Oliveri et al., 2018). All these aspects should be properly commented in the discussion section, comparing the different results, evidencing eventual differences and briefly resuming a mechanistic role for miR-21 in all the processes relevant for organ fibrosis and diabetes complications in general, another largely discussed topics. Also, the authors should evidence the novelty of their findings.
The passage introducing Nox 2 is not clear. Why this protein was studied? It is a target of miR-21. Why have you looked specifically at this protein? More justification should be provided.
The abstract is too long. It should be shortened a presented in a more focused manner.
Technical description of methods is sufficiently detailed.
Minor points:
There is a an empty conclusion section after the methods section.
reference list is doubled (numbers are reported twice).
Author Response
Reviewer 2:
These findings are not novel since a key role for mir 21 in diabetic wounds has been shown (ref 14-18), as has a role in macrophage polarization (Xi et al 2018 oncogene, Wang et al., 2015 Plos one; Sahraei et al., 2019 JCI). Similarly, the increase of miR-21 in the diabetic envirnoment (human, mouse, intracellular and extracellular) has been largely described (Prattichizzo et al., 2018 Redox Biol; Oliveri et al., 2018). All these aspects should be properly commented in the discussion section, comparing the different results, evidencing eventual differences and briefly resuming a mechanistic role for miR-21 in all the processes relevant for organ fibrosis and diabetes complications in general, another largely discussed topics. Also, the authors should evidence the novelty of their findings.
- Thanks for the comments. We added to discussion regarding these papers (line 225 to 230). For the novelty of our paper you can also find in the discussion. We think that our paper is the first description which divided miR-21 expression in 3 phase during diabetic wound healing. Also the pathway HG/miR-21/PI3K/NOX2/ROS we identified is the first to connect miR-21 to diabetic wound phenotype.
The passage introducing Nox 2 is not clear. Why this protein was studied? It is a target of miR-21. Why have you looked specifically at this protein? More justification should be provided.
- We agree. We added more information in the passage to justify the reason we worked on NOX2 gene in the introduction section (line 63 to 67).
The abstract is too long. It should be shortened a presented in a more focused manner.
- We shorten the abstract to be more focused.
Minor points:
There is an empty conclusion section after the methods section.
- The empty conclusion was removed.
reference list is doubled (numbers are reported twice).
- This has been fixed.

Reviewer 3 Report
Summary
The Authors hypothesized that miR-21 plays a role in regulating inflammation by promoting M1 macrophage polarization and the production of ROS. They demonstrate that the time-course of changes in miR-21 abundance during wound repair between diabetic and no diabetic animals. In non-diabetic animals miR-21 levels peaked at 7-days and then began to descend whereas in diabetic animals the rise was slower and continued to increase throughout the 21-d term of the experiment. miR-21 was more abundant when RAW 264.7 macrophages were induced into the M1 phenotype than the M2 phenotype. Overexpression of miR-21 in macrophage cells produced a modest increase in the abundance of IL-1B, TNFalpha, iNOS, and IL-6 mRNA – markers of the M1 phenotype. Culturing RAW 264.7 macrophages in the presence of 25 mM glucose produced a modest, time-dependent increase in the abundance of miR-21 and a reduction in the abundance of PTEN mRNA. The authors also report that transfecting RAW 264.7 macrophages with a miR-21 mimic increased NOX2 expression and ROS production as well as the abundance of immunoreactivity for phosphorylated ERK1/2, phosphorylated PI 3-kinase, and NOX2. Finally, inhibiting PI 3-kinase activity blocked the ability of miR-21 overexpressing to increase the abundance of NOX2 mRNA. The authors conclude that miR-21 is involved in promoting polarization of macrophage to the M1 phenotype and that dysregulation of miR-21 in diabetic wounds may explain the abnormal inflammation and persistent M1 macrophages observed in diabetic wounds.
Critique
General – The manuscript would benefit from being revised for English grammar including use of the correct tense. Throughout this manuscripts, the authors mistakenly refer to changes in mRNA abundance as changes in gene expression. This is incorrect. mRNA abundance is affected by changes in both transcription and stability. The author’s data suggest any involvement of miR-21 in converting RAW 264.7 macrophages to, or maintaining them in, the M1 phenotype is weak at best. Figure 2 shows that INF-r/LPS produced a 25-fold increase in miR-21 and large increases in IL-1b, TNFalpha, iNOS, and IL-6 mRNA whereas the transfection experiments with miR-21 mimic increased miR-21 over 1000-fold but had only modest effects on the abundance of IL-1b, TNFalpha, iNOS, and IL-6 mRNA.
Although the abstract states that hyperglycemia increased NOX2 expression and ROS production, this data was not included in the manuscript. Figure 5 panels C and D showed that transfection with a miR-21 mimic increased ROS production and NOX mRNA, respectively.
Figure 2 title. The authors are not examining gene expression, they are measuring the abundance of mRNA in their macrophage populations.
Line 145: “that miR-21 gene expression is increased”
The assay does not examine gene expression, miR-21 abundance is assessed.
Line 147 – qPCR does not measure gene expression
Figure 4. The effects of transfecting miR-21 into RAW macrophages, resulting in a 1000-fold increase in its abundance, are modest at best when assessing mRNAs that are indicators of the M1 phenotype as shown in Figure 2.
Line 148: figure 4B is being referred to as figure 3B in this line of text.
Line 165: are PTEN transcripts the only know target of miR-21?
Line 166. The authors are not looking at gene expression.
Figure 5, panel A: was the effect of 4 h incubation in 25 mM glucose on miR-21 levels significant relative to the 5 mM glucose controls?
Line 199: the authors are examining changes in ERK1/2 phosphorylation by immunoblotting – they are no looking at pERK1/2 expression.
Figure 6A. The data shown for total and phospho PI3K in the representative images is not at all convincing that there is an increase in PI3K phosphorylation. In fact, by eye, there appears to be a decrease in ‘total’ PI3K immunoreactivity.
Lines 257-258. “miR-21 repressed miR-21 target gene 257 pTEN”
25 mM glucose increased the abundance of miR-21 and reduced the abundance of PTEN mRNA (Figure 5B). The authors failed to establish a causal relationship and they did not examine the involvement of miR-21 in hyperglycemia-induced reduction in PTEN mRNA.
Reviewer 4 Report
The manuscript by Liechty et al. titled “Role of microRNA-21 and its underlying mechanisms in inflammatory responses in diabetic wounds” is an intriguing investigation. The study provides new findings to how miR-21 acts on the inflammatory condition present in diabetic wound through influencing macrophage polarization and ROS production. In general, the results support their conclusion. I have some comments that I believe might help the authors in increasing the impact of this manuscript. Author should show an immunohistochemistry analysis of the wound skin in term of miR-21 expression and macrophage content at the different time points following wounding. Moreover, wound macrophages isolation (for example employing a polyvinyl alcohol (PVA)-sponge implantation approach) could be useful to better characterized the different macrophages present in the wound skin. Lastly, regarding written form, there are some mistakes and I suggest the revision of the manuscript by a technical writer.
Author Response
The manuscript by Liechty et al. titled “Role of microRNA-21 and its underlying mechanisms in inflammatory responses in diabetic wounds” is an intriguing investigation. The study provides new findings to how miR-21 acts on the inflammatory condition present in diabetic wound through influencing macrophage polarization and ROS production. In general, the results support their conclusion.
- We thank the reviewer’s encouraging comments. We feel that our work is novel and contributes significantly to the field in diabetic wound healing and in general.
Author should show an immunohistochemistry analysis of the wound skin in term of miR-21 expression and macrophage content at the different time points following wounding.
- This is a great point. In our study, we compared miR-21 expression between non-diabetic wounds and diabetic wounds and found that miR-21 was differentially expressed among these conditions. More interestingly, its expression can be classified into 3 phases: miR-21 was higher in diabetic wounds at day 1 after injury (inflammatory phase), then at day 3 and day 7 after injury (proliferative phase), miR-21 was lower in diabetic wounds, then at day 14 and 21 after injury (remodeling phase), miR-21 was significantly higher in diabetic wounds. Furthermore, we confirmed that miR-21 was higher on day 1 after wounding in diabetic wounds and human diabetic skin. The data indicates miR-21 may play different roles in each phase of the wound healing process, also implying the importance of time on the modulation of miR-21 expression in its target therapy. Our data also provided a potential solution to the controversy in miR-21-based therapy. We agree if we have IHC data for miR-21 expression during wound healing, it will make our paper more informative. We will study this and report it in our future papers.
Moreover, wound macrophages isolation (for example employing a polyvinyl alcohol (PVA)-sponge implantation approach) could be useful to better characterize the different macrophages present in the wound skin.
- We understand the importance of wound macrophage during wound healing process. We isolated non-Db and Db wound macrophage, did miR-21 expression analysis, and found that miR-21 was significantly higher in diabetic wound macrophage at day 1 after injury. This data was consistent with its expression in wounds. We added this data to Figure 1, panel C.
Lastly, regarding written form, there are some mistakes and I suggest the revision of the manuscript by a technical writer.
- We revised the manuscript to improve its clarity.
Round 2
Reviewer 3 Report
In the revised version of the manuscript, the Authors have made extensive corrections to the text and this has greatly improved its readability.
Although the Authors have made a serious effort to revise this manuscript, they have not actually addressed the weaknesses pointed out in my review of the previous version.
The Authors conclude the abstract by stating: Dysregulation of miR-21 may explain the abnormal inflammation and persistent M1 macrophage polarization seen in diabetic wounds.
As stated in my review of the previous version of this manuscript, the Authors’ own data shows that if miR-21 is involved in macrophage polarization, its role is a minor one.
In my previous review I also pointed out that the assessment of PI3K shown in Figure 6, the increase in the ratio of phospho to total PI3K appeared to be due to a decrease in total PI3K immunoreactivity rather than an increase in phospho PI3K. Hence, there has not been an increase in abundance of phospho PI3K in these cells. The ratio is altered due to the reduction in total PI3K not an increase in phospho PI3K.
In my previous review, I indicated 25 mM glucose increased the abundance of miR-21 and reduced the abundance of PTEN mRNA (Figure 5B). The authors failed to establish a causal relationship and they did not examine the involvement of miR-21 in hyperglycemia-induced reduction in PTEN mRNA.,
The Authors have not shown that the increase in miR-21 produced by 25 mM glucose caused the reduction in PTEN mRNA. Their data simply shows that hyperglycemia increased miR-21 and decreased PTEN mRNA.
Note – miRs do not bind to genes. They interact with mRNA. Furthermore, inhibition of PTEN does not ACTIVATE PI3K. PTEN is a lipid phosphatase which removes the phosphoryl group from the D3 position of the inositol ring of phosphatidylinositol 3,4,5-trisphosphate, phosphatidylinositol 3,4-diphosphate, phosphatidylinositol 3-phosphate, and inositol 1,3,4,5-tetrakisphosphate. Inhibition of PTEN, the enzyme that dephosphorylates the 3 position of phosphoinositides (if they are, in fact, phosphorylated at position 3 of the inositol ring), would result in an increase in 3-phosphorylated phosphoinositides and hence increase the activation of signaling downstream of PI3K
Figure 2 - 4. INF-r/LPS produced a 25-fold increase in miR-21 and large increases in IL-1b, TNFalpha, iNOS, and IL-6 mRNA whereas the transfection experiments with miR-21 mimic increased miR-21 over 1000-fold but had only modest effects on the abundance of IL-1b, TNFalpha, iNOS, and IL-6 mRNA. The authors have responded to this comment by stating, quite possible correctly, that induction of the inflammatory makers by LPS or miR-21 may through different mechanisms. This does not alter their own observations that a 1000-fold increase in miR-21 resulted in a small increase in the abundance of transcripts that were chosen as markers of the M1 macrophage phenotype – in other words, this large increase in miR-21 had minimal effects on macrophage polarization. The actual title of the paper states that miR-21 regulates macrophage polarization and ROS production.
